# The role of interleukin-6 as a prognostic biomarker for predicting acute exacerbation in interstitial lung diseases

**Jae Ha Lee**[1], **Ji Hoon Jang**[1], **Jin Han Park**[1], **Hang-Jea Jang**[1], **Chan Sun Park**[2], **Sunggun Lee**[3], **Seong-Ho Kim**[3], **Ji Yeon Kim**[4], **Hyun Kuk Kim**[1]*

1 Division of Pulmonology, Department of Internal Medicine, Inje University Haeundae Paik Hospital, Inje University College of Medicine, Busan, Republic of Korea, 2 Division of Allergy, Department of Internal Medicine, Inje University Haeundae Paik Hospital, Inje University College of Medicine, Busan, Republic of Korea, 3 Division of Rheumatology, Department of Internal Medicine, Inje University Haeundae Paik Hospital, Inje University College of Medicine, Busan, Republic of Korea, 4 Division of Pathology, Inje University Haeundae Paik Hospital, Inje University College of Medicine, Busan, Republic of Korea

* khkmd1205@hanmail.net

## Abstract

### Background

Interstitial lung diseases (ILDs) are chronic, parenchymal lung diseases with a variable clinical course and a poor prognosis. Within various clinical courses, acute exacerbation (AE) is a devastating condition with significant morbidity and high mortality. The aim of this study was to investigate the role of interleukin-6 (IL-6) to predict AE and prognosis in patients with ILD.

### Methods

Eighty-three patients who were diagnosed with ILD from 2016 to 2019 at the Haeundae Paik Hospital, Busan, South Korea, were included and their clinical data were retrospectively analyzed.

### Results

The median follow-up period was 20 months. The mean age was 68.1 years and 65.1% of the patients were men with 60.2% of patients being ever-smokers. Among ILDs, idiopathic pulmonary fibrosis was the most common disease (68.7%), followed by connective tissue disease-associated ILD (14.5%), cryptogenic organizing pneumonia (9.6%), and nonspecific interstitial pneumonia (6.0%). The serum levels of IL-6 were measured at diagnosis with ILD and sequentially at follow-up visits. During the follow-ups, 15 (18.1%) patients experienced an acute exacerbation (AE) of ILD and among them, four (26.7%) patients died. In the multivariable analysis, high levels of IL-6 (OR 1.014, 95% CI: 1.001–1.027, p = 0.036) along with lower baseline saturations of peripheral oxygen ($SpO_2$) were independent risk factors for AE. In the receiver operating characteristic curve analysis, the area under the curve was 0.815 ($p < 0.001$) and the optimal cut-off value of serum IL-6 to predict AE was 25.20 pg/mL with a sensitivity of 66.7% and specificity of 80.6%. In the multivariable Cox

**Data Availability Statement:** All relevant data are within the manuscript and its Supporting information files.

**Funding:** This work was supported by 2019 Inje university research grant (20190041). The grant did not have any influence on the research results and was a support to promote the research activities.

**Competing interests:** The authors have declared that no competing interests exist.

analysis, a high level of serum IL-6 (HR 1.007, 95% CI: 1.001–1.014, p = 0.018) was only an independent risk factor for mortality in ILD patients.

## Conclusions

In our study, a high level of serum IL-6 is a useful biomarker to predict AE and poor prognosis in patients with ILD.

## Introduction

Interstitial lung diseases (ILD) are a heterogeneous group of diffuse parenchymal lung disorders with highly variable clinical courses and poor outcomes [1]. Within variable clinical courses, acute exacerbation (AE) is well-known as a life threatening condition with significant morbidity and high mortality [2, 3]. In terms of AE, the incidence in patients with idiopathic pulmonary fibrosis (IPF) is 5–10% per year with a median survival of less than 3 months [4–6].

Previous studies reported that old age, lower lung function including forced vital capacity (FVC) and diffusing capacity of the lung for carbon monoxide (DLco), and distances or de-saturation during the six minute walk test (6MWT) were risk factors for AE IPF [7, 8]. However, because of limitations of physiologic parameters such as dependency on patient efforts or interobserver variability, predicting AE remains difficult [9].

Serum biomarkers are relatively easy to measure independently of patient effort or observer ability. Within the pathogenesis of AE-IPF, there have been several reports that cytokines may play an important role [10, 11]. Among cytokines, IL-6 is a soluble mediator with pleiotropic effects on inflammation, immune responses, and fibrosis [12, 13]. In a recent study, Shochet et al. reported in experimental research between IPF patients and normal healthy donors that IL-6 trans-signaling components lead to indirect TGF-β, which is well-known as a pro-fibrotic growth factor with an influence on pathway activation and disease progression, suggesting the importance of IL-6 in IPF pathogenesis [14]. Therefore, our aim in this study was to evaluate the role of IL-6 as a biomarker for predicting AE and prognosis in patients with ILD.

## Materials and methods

### Study subjects

Eighty-three patients who were diagnosed with ILD at the Haeundae-Paik Hospital, Busan, Republic of Korea, from December 2016 to September 2019, were included in this study. Among the patients with ILD, patients who had serum IL-6 levels measured at diagnosis with ILD and at follow-up visits consecutively every 2–3 months were included. All the patients met the diagnostic criteria in the international guidelines set by the American Thoracic Society (ATS) and European Respiratory Society (ERS) [15, 16]. This study was approved by the Institutional Review Board of the Haeundae-Paik Hospital (approval number: 2019-12-036), and the requirement for written informed consent was waived due to the retrospective nature of this study.

### Measurement of IL-6

IL-6 samples were taken at the diagnosis with ILD and sequentially at follow-up visits every 2–3 months. Serum IL-6 concentrations (pg/mL) were measured though an

electrochemiluminescence immunoassay (ECLIA) using the Cobas e411 analyzer (Hitachi High-Technologies, Japan).

## Clinical information

Clinical data were retrospectively obtained from the medical records at January 3, 2020. Pulmonary function testing, a measurement of the diffusing capacity of the lung for carbon monoxide (DLco), forced expiratory volume in one second, and forced vital capacity (FVC) were performed according to the recommendations of the ATS/ERS [17–19]. The results were expressed as percentages of normal predicted values. The 6-minute walk test was performed according to ATS guidelines [20]. Data from the diagnosis to the follow-up clinic visits, which were conducted normally every 2–3 months, and those from hospitalizations were reviewed to determine the development of AE. The diagnostic criteria for AE-ILD, based on Collard et al., were as follows: 1) previous or concurrent IPF diagnosis; 2) acute worsening or development of dyspnea typically within the past month; 3) high-resolution computed tomography (HRCT) with new bilateral ground-glass opacity and/or consolidation superimposed on a background pattern consistent with typical interstitial pneumonia patterns; and 4) deterioration not fully explained by cardiac failure or fluid overload [21].

## Statistical analysis

The data presents the frequency and percentage for categorical variables and mean ± standard deviation (SD) for numeric variables. Differences in study participants' characteristics were compared across subgroups with a chi-square test or Fisher's exact test for categorical variables and an independent test or Mann-Whitney's U test for continuous variables as appropriate. To determine if its distribution was normal, we used the Shapiro-Wilk's test. Univariate and multivariate analyses, using logistic regression, were performed in order to identify prognostic factors which are independently related to AE. The receiver operating characteristic (ROC) curve analysis was performed to assess the sensitivity and specificity of IL-6 for predicting AE. Overall survival (OS) was estimated using the Kaplan-Meier curve. Survival curves were compared between the groups using the log-rank test. Multivariate analyses, using Cox regression with backward and stepwise elimination, were performed in order to identify prognostic factors which are independently related to mortality. All statistical analyses were carried out using SPSS 24.0 (IBM Corp, Armonk, USA), and p values less than 0.05 were considered statistically significant.

# Results

## Study population

From December 2016 to September 2019, 405 patients with ILD at Haeundae Paik hospital (Busan, Republic of Korea) were screening for this study. Among them, patients who did not perform IL-6 test or follow up at IL-6 test at least 6 months were excluded (Fig 1). A total of 83 patients with ILD were included in this study. The median follow-up period was 20 months. The mean age of the study population was 68.1 years with 65.1% male patients and 60.2% of patients being ever-smokers. Among ILDs, IPF was the most common (68.7%), followed by connective tissue disease–associated (CTD) ILD (14.5%), cryptogenic organizing pneumonia (9.6%), and nonspecific interstitial pneumonia (6.0%) (Table 1).

## Baseline characteristics

Most patients exhibited a mild restrictive ventilator defect and reduced DLco. In 6MWT, the mean baseline and minimum $SpO_2$ was 95.9% and 90.6%, respectively. The mean baseline

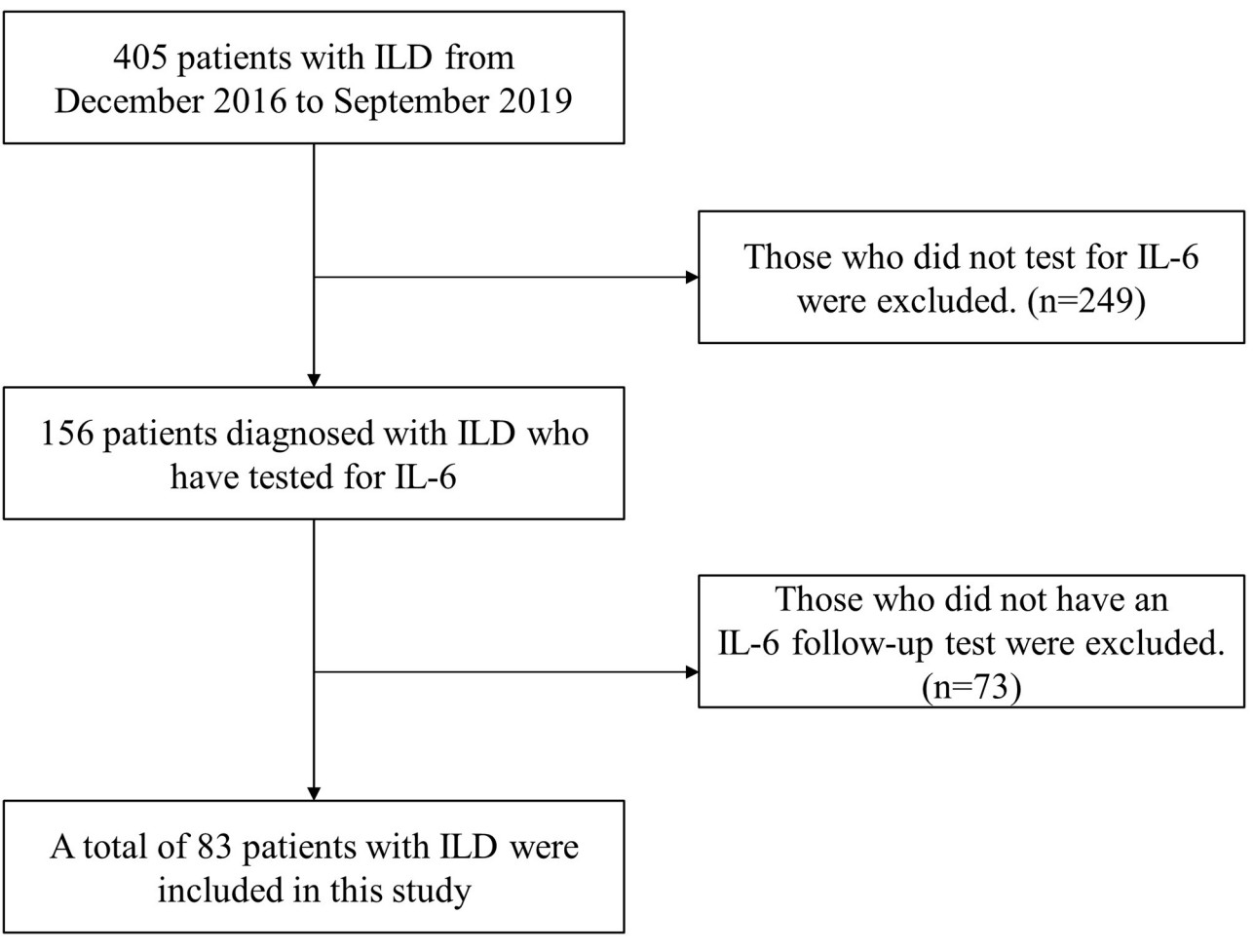

**Fig 1. Flow chart of patients enrollment.** ILD, interstitial lung disease; IL-6, interleukin-6.

level of serum of IL-6 was 14.9 pg/mL. During the follow-up period, 15 patients (18.1%) suffered AE. The AE group exhibited higher levels of C-reactive proteins (CRP) and white blood cells (WBC) with lower levels of baseline $SpO_2$ compared with the no-AE group (Table 2).

### Predictive factors for AE

Among the 15 (18.1%) patients with AE (idiopathic AE: 11 patients and triggered AE: 4 patients), IPF patients were most common (80%) while the other patients were classified with CTD (20%). In the univariable logistic regression analysis, high levels of serum IL-6 during the follow-up period, CRP, white blood cell (WBC) count, lactate dehydrogenase (LDH), baseline $SpO_2$, and disease duration were significant predictive factors for AE in patients with ILD (Table 3).

In the multivariable analysis, high levels of serum IL-6 (OR 1.014, 95% CI: 1.001–1.027, p = 0.036) during the follow-up period and lower baseline $SpO_2$ (OR 0.750, 95% CI: 0.568–0.991, p = 0.043) were independent predictive factors for AE. In the receiver operation characteristics curves analysis, high levels of serum IL-6 were useful in predicting for AE (area under the curve = 0.815, 95% CI: 0.704–0.927, p < 0.001) in patients with ILD. The best cut-off level was 25.20 pg/mL with a sensitivity of 66.7% and specificity of 80.6% (Fig 2).

**Table 1. Baseline clinical characteristics of the patients.**

| Characteristics | All Patients (n = 83) |
|---|---|
| Age, years | 68.1 ± 10.5 |
| Male, n (%) | 54 (65.1) |
| Ever smokers, n (%) | 50 (60.2) |
| Height, cm | 162.3 ± 8.5 |
| Weight, kg | 64.7 ± 10.9 |
| BMI, kg/m$^2$ | 24.5 ± 3.4 |
| Home O$_2$, n (%) | 5 (6.0) |
| Interstitial lung disease | |
| IPF, n (%) | 57 (68.7) |
| CTD, n (%) | 12 (14.5) |
| COP, n (%) | 8 (9.6) |
| NSIP, n (%) | 5 (6.0) |
| HP, n (%) | 1 (1.2) |
| Radiologic pattern | |
| UIP, n (%) | 56 (67.4) |
| NSIP, n (%) | 10 (12.1) |
| OP, n (%) | 10 (12.1) |
| Indeterminate, n (%) | 6 (7.2) |
| Pulmonary function | |
| FVC, % predicted | 73.9 ± 15.6 |
| DLco, % predicted | 59.1 ± 17.0 |
| TLC, % predicted | 72.5 ± 12.4 |
| FEV1/FVC, % | 77.0 ± 8.5 |
| Six-minute walk test | |
| Distance, m | 402.2 ± 99.0 |
| Baseline SpO$_2$, % | 95.9 ± 2.5 |
| Lowest SpO$_2$, % | 90.6 ± 5.9 |
| BAL | |
| Total WBC count, cell/μL | 2,135.7 ± 3,402.5 |
| Neutrophil, % | 28.6 ± 26.3 |
| Lymphocyte, % | 18.6 ± 21.8 |
| PaO$_2$, mmHg | 88.0 ± 28.2 |
| BNP, pg/mL | 421.1 ± 734.5 |
| Baseline IL-6, pg/mL | 14.9 ± 33.8 |
| CRP, mg/dL | 2.5 ± 5.5 |
| WBC, cell/μL | 8,094.8 ± 3,041.7 |
| LDH, IU/L | 264.8 ± 86.3 |

Values are presented as mean ± standard deviation or number (%).

BMI, body mass index; IPF, idiopathic pulmonary fibrosis; CTD, connective tissue disease-associated interstitial lung disease; COP, cryptogenic organizing pneumonia; NSIP, non-specific interstitial pneumonia; HP, hypersensitivity pneumonitis; UIP, usual interstitial pneumonia; OP, organizing pneumonia; FVC, forced vital capacity; DLco, diffusing capacity of the lungs for carbon monoxide; TLC, total lung capacity; SpO2, saturation of peripheral oxygen; BAL, bronchoalveolar lavage; WBC, white blood cell; PaO2, partial pressure of oxygen; BNP, brain natriuretic peptide; IL-6, interleukin-6; CRP, C-reactive protein; LDH, lactate dehydrogenase.

**Table 2. Comparison of characteristics between ILD patients with and without AE.**

| Variable | AE (+) | No AE(-) | P-value |
|---|---|---|---|
| | (n = 15) | (n = 68) | |
| Number of patients, n (%) | 15 (18.1) | 68 (81.9) | |
| Age, years | 70.3 ± 5.8 | 67.6 ± 11.2 | .477 |
| Male, n (%) | 10 (66.7) | 44 (64.7) | .885 |
| Ever smokers, n (%) | 7 (46.7) | 43 (64.2) | .209 |
| BMI, kg/m$^2$ | 23.9 ± 4.1 | 24.6 ± 3.3 | .368 |
| Interstitial lung disease | | | |
| IPF, n (%) | 12 (80.0) | 45 (66.2) | .296 |
| CTD, n (%) | 3 (20.0) | 9 (13.2) | .500 |
| Pulmonary function | | | |
| FVC, % predicted | 69.5 ± 13.1 | 74.9 ± 15.9 | .219 |
| DLco, % predicted | 52.5 ± 16.8 | 60.5 ± 16.8 | .179 |
| TLC, % predicted | 76.0 ± 13.4 | 71.6 ± 12.4 | .536 |
| Six-minute walk test | | | |
| Distance, m | 355.9 ± 101.9 | 410.8 ± 96.8 | .084 |
| Baseline SpO$_2$, % | 94.3 ± 2.3 | 96.2 ±2.4 | .009 |
| Lowest SpO$_2$, % | 88.8 ± 5.5 | 90.9 ± 5.9 | .153 |
| Baseline IL-6, pg/mL | 35.5 ± 68.3 | 10.5 ± 17.5 | .057 |
| Peak IL-6*, pg/mL | 133.5 ± 205.9 | 21.1 ± 35.0 | < .001 |
| CRP, mg/dL | 7.9 ± 9.9 | 1.2 ± 2.5 | < .001 |
| WBC, cell/μL | 9,614.0 ± 4,169.3 | 7,759.7 ± 2,657.2 | .024 |
| LDH, IU/L | 317.0 ± 147.7 | 255.1 ± 67.0 | .226 |

Values are presented as mean ± standard deviation or number (%).

AE, acute exacerbation; BMI, body mass index; IPF, idiopathic pulmonary fibrosis; CTD, connective tissue disease-associated interstitial lung disease; FVC, forced vital capacity; DLco, diffusing capacity of the lungs for carbon monoxide; TLC, total lung capacity; SpO2, saturation of peripheral oxygen; IL-6, interleukin-6; CRP, C-reactive protein; WBC, white blood cell; LDH, Lactate dehydrogenase.

*Peak IL-6 –highest serum level of IL-6 among sequentially measured IL-6 during follow-up.

## Survival analysis

During the follow-up period, four patients (4.8%) died. All deaths occurred in the AE groups and all causes of mortality were AE ILD. In the AE groups, 2-year and 3-year survival rates were 92.3% and 65.9%, respectively (S1 Fig). In a comparison of survival between high and low levels of IL-6 groups, the survival rate was significantly high in the low IL-6 group (Log rank test, p = 0.018) (Fig 3).

In a univariable Cox analysis, older age, high levels of IL-6, and shorter distances during 6MWT were significant prognostic factors for mortality. However, in multivariable Cox analysis, only high levels of IL-6 were found to be a significant factor affecting overall survival (HR 1.007, 95% CI: 1.001–1.014, p = 0.018) (Table 4).

## Discussion

In our study, high levels of serum IL-6 during the follow-up period and lower baseline levels of SpO$_2$ were independent predictive factors for AE in patients with ILD. The best cut-off level of IL-6 to predict AE was 25.20 pg/mL. In addition, high levels of IL-6 during the follow-up period were an independent prognostic factor for mortality in patients with ILD.

**Table 3. Predictive factors for AE in ILD patients assessed using the logistic regression model.**

| Variable | Univariate Analysis | | Multivariate Analysis | |
|---|---|---|---|---|
| | OR (95% CI) | *P*-value | OR (95% CI) | *P*-value |
| Age | 1.028 (0.969–1.091) | .362 | | |
| Male | 1.091 (0.034–3.561) | .885 | | |
| Ever smokers | 0.488 (0.158–1.513) | .214 | | |
| BMI | 0.942 (0.791–1.122) | .501 | | |
| Disease duration | 1.031 (1.005–1.057) | .017 | | |
| Interstitial lung disease | | | | |
| IPF | 0.489 (0.125–1.908) | .303 | | |
| CTD | 0.610 (0.144–2.592) | .503 | | |
| Radiologic pattern | | | | |
| UIP | 0.115 (0.014–0.930) | .430 | | |
| Pulmonary function | | | | |
| FVC | 0.995 (0.940–1.014) | .218 | | |
| DLco | 0.969 (0.934–1.006) | .104 | | |
| TLC | 1.032 (0.939–1.134) | .516 | | |
| Six-minute walk test | | | | |
| Distance | 0.995 (0.989–1.001) | .084 | | |
| Baseline SpO$_2$ | 0.738 (0.573–0.952) | .019 | 0.750 (0.568–0.991) | .043 |
| Lowest SpO$_2$ | 0.948 (0.862–1.042) | .265 | | |
| PaO$_2$ | 0.994 (0.969–1.019) | .628 | | |
| Baseline IL-6 | 1.018 (0.999–1.037) | .062 | | |
| *Peak IL-6 | 1.017 (1.006–1.028) | .002 | 1.014 (1.001–1.027) | .036 |
| CRP | 1.223 (1.054–1.442) | .009 | | |
| WBC | 1.000 (1.000–1.000) | .045 | | |
| LDH | 1.007 (1.000–1.013) | .049 | | |

OR: odds ratio; CI: confidence interval; BMI, body mass index; IPF, idiopathic pulmonary fibrosis; CTD, connective tissue disease-associated interstitial lung disease; UIP, usual interstitial pneumonia; FVC, forced vital capacity; DLco, diffusing capacity of the lungs for carbon monoxide; TLC, total lung capacity; SpO2, saturation of peripheral oxygen; PaO$_2$, partial pressure of oxygen; IL-6, interleukin-6; CRP, C-reactive protein; WBC, white blood cell; LDH, lactate dehydrogenase.

*Peak IL-6 –highest serum level of IL-6 among sequentially measured IL-6 during follow-up.

Within heterogeneous and diverse clinical courses of ILD, AE is a clinically significant event resulting high mortality. Despite the efforts of many researchers, the actual pathogenesis of AE is not well-known and the mortality of AE remains high. Recently, there have been increasing reports about plasma biomarkers that may be useful for predicting AE in patients with ILD in the pathobiology of AE ILD. Among them, previous studies reported that IL-6 also may be a useful predictor of AE [22, 23]. Firstly, IL-6 is considered a pro-inflammatory cytokine in the acute phase of inflammation, which is produced by various stimuli as well as a promotor specific immune response [24]. In a previous study, the authors suggested that IL-6, as a systemic pro-inflammatory cytokine, played an important role in the innate cells of the pathogenesis of IPF [11]. In addition to the role of IL-6 as a pro-inflammatory cytokine, there were some reports that IL-6 is associated with a fibrotic response and can promote fibrosis by driving chronic inflammation and activating the TGF-β pathway [25]. In a rat model of experimental fibrosis, IL-6 concentrations were shown to be increased in rats and were associated with a proliferative response in fibroblasts [12, 13]. Although the actual role of IL-6 in AE ILD

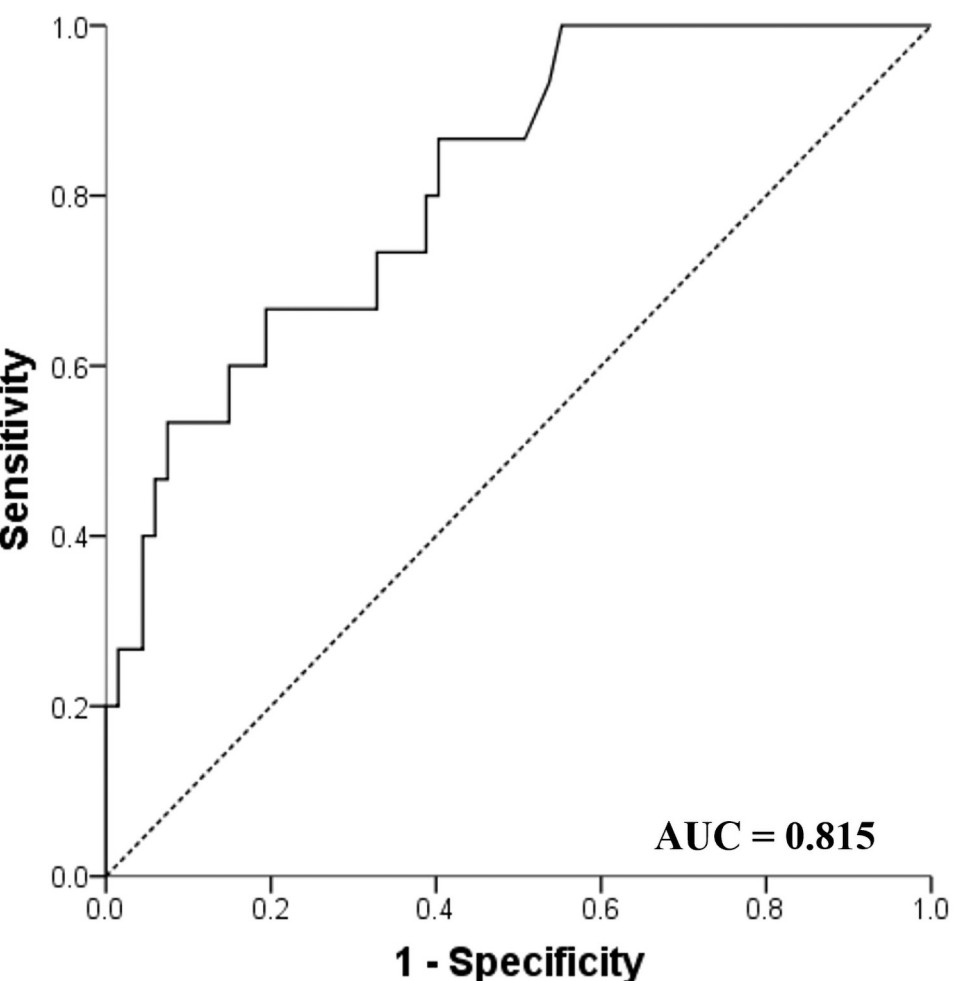

**Fig 2. Receiver Operating Characteristic (ROC) curve of IL-6 to predict AE.** IL-6, interleukin 6.

remains elusive, we assumed that IL-6 could be an important plasma biomarker because IL-6 possesses both the nature of pro-inflammatory and pro-fibrotic mediators. Further studies must be performed in order to elucidate the role of IL-6 in patients with AE ILD.

Our study showed that impairment of oxygenation in the resting state was a useful predictor of AE ILD. A previous study also supported our results in this study. In 1,019 patients with ILD (AE group 193, non-AE group 826), Suzuki et al. found that lower levels of partial pressure of oxygen ($P_aO_2$) in arterial blood gas in the resting state was an independent risk factor for AE (HR = 0.98, 95% CI: 0.97–0.99, p = 0.013) [26]. In the analysis of INPULSIS trial data to investigate risk factors for AE, Collard et al. reported that among the patients with investigator-reported AE, the use of baseline supplemental oxygen was an independent risk factor for AE (HR = 2.47, 95% CI: 1.37–4.47, p = 0.001) [27]. In 108 patients with IPF, a previous study reported similar results with our study. Okuda et al. showed that the minimum $SpO_2$ level of 88% or less during 6MW was a predictive factor for predicting AE (HR 0.86, 95% CI: 0.80–0.93, p < 0.001) [28]. Therefore, when low baseline levels of $SpO_2$ were observed in addition to the minimal level of $SpO_2$ during 6MWT, we needed to also consider low baseline levels of $SpO_2$ as a risk factor for AE.

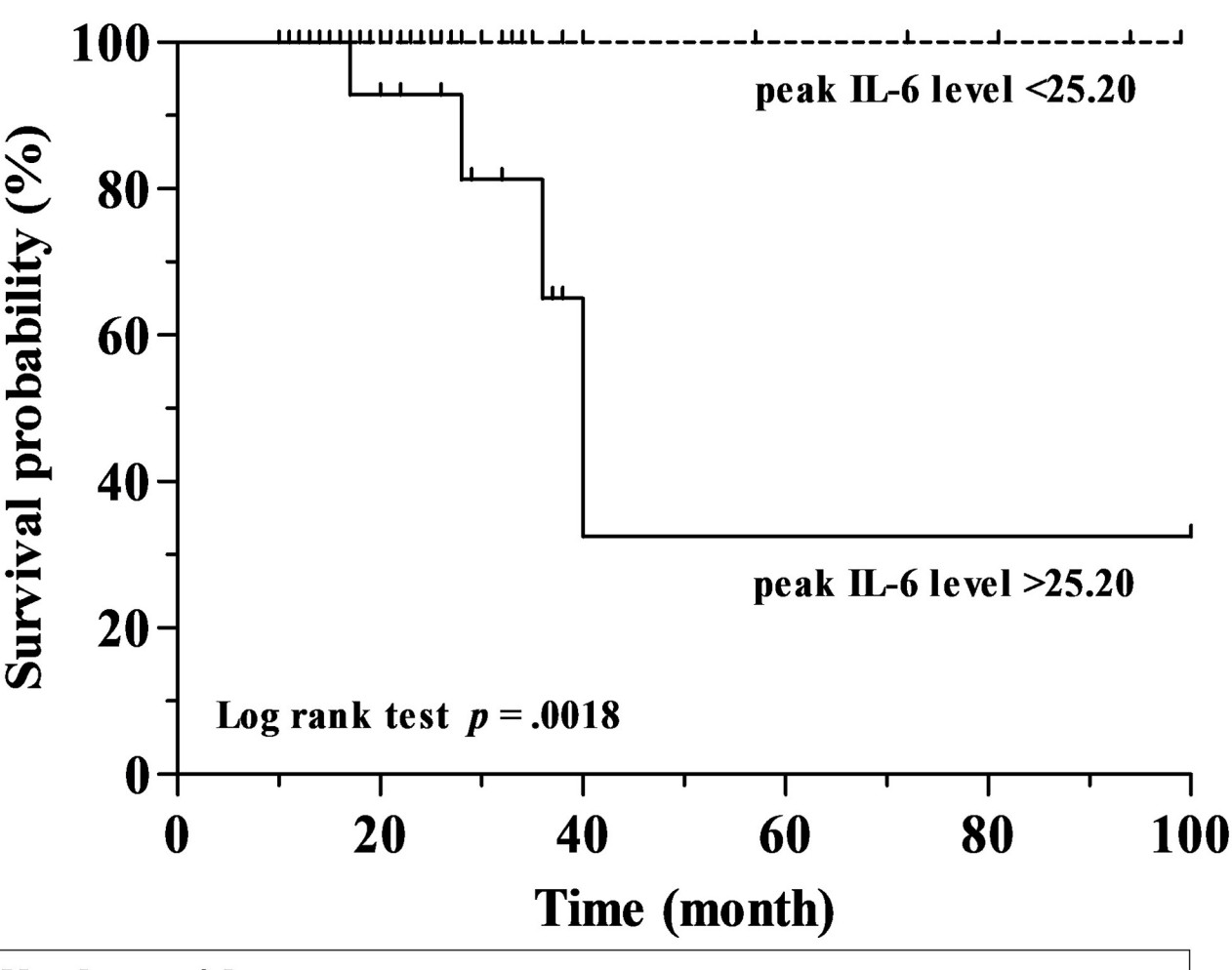

| Number at risk | | | | | | |
|---|---|---|---|---|---|---|
| | 0 | 20 | 30 | 40 | 60 | 100 |
| peak IL-6 level <25.20 | 60 | 60 | 60 | 60 | 60 | 60 |
| peak IL-6 level >25.20 | 23 | 22 | 21 | 19 | 19 | 19 |

**Fig 3. Comparison of Kaplan–Meier survival curves between high and low levels of IL-6.** IL-6, interleukin 6.

In our study, high levels of IL-6 at sequential measurements during the follow-up period was an independent risk factor for AE in patients with ILD. Recently in 41 patients with IPF, Spyros et al. reported that high levels of IL-6 characterized the early diagnosis of AE-IPF (IL-6 between the AE-IPF group vs stable IPF group, 6.2 pg/mL vs 2.1pg/mL, p = 0.002). The authors suggested that the role of pro-inflammatory and pro-fibrotic markers on IL-6 may play an important factor and further studies are necessary to clarify the enigma of AE pathogenesis [22]. Also, in our study, a high serum IL-6 level of 25.20 pg/mL was the most discriminatory optimal cut-off value predicting AE in patients with ILD. So far, the optimal cut-off value of IL-6 to predict AE has not been determined. Therefore, in cases of high levels of IL-6 above 25.20 pg/mL in patients with ILD, it is necessary to suspect AE.

**Table 4. Prognostic factors for mortality in patients with ILD assessed using cox proportional hazards model.**

| Variable | Univariate Analysis | | Multivariate Analysis | |
|---|---|---|---|---|
| | HR (95% CI) | P-value | HR (95% CI) | P-value |
| Age | 1.211 (1.002–1.465) | .048 | | |
| Male | 0.206 (0.021–1.993) | .172 | | |
| Ever smokers | 0.016 (0.000–51.983) | .317 | | |
| BMI | 0.746 (0.504–1.104) | .143 | | |
| Disease duration | 0.810 (0.581–1.130) | .215 | | |
| Interstitial lung disease | | | | |
| IPF | 0.667 (0.068–6.505) | .728 | | |
| CTD | 0.857 (0.089–8.294) | .894 | | |
| Radiologic pattern | | | | |
| UIP | 0.031 (0.001–598.0) | .491 | | |
| Pulmonary function | | | | |
| FVC | 1.031 (0.936–1.135) | .538 | | |
| DLco | 1.011 (0.946–1.081) | .742 | | |
| TLC | 1.177 (0.903–1.535) | .229 | | |
| Six-minute walk test | | | | |
| Distance | 0.988 (0.976–0.999) | .037 | | |
| Baseline SpO$_2$ | 0.802 (0.502–1.280) | .355 | | |
| Lowest SpO$_2$ | 0.992 (0.809–1.217) | .938 | | |
| PaO$_2$ | 0.998 (0.929–1.072) | .955 | | |
| Baseline IL-6 | 0.992 (0.953–1.033) | .694 | | |
| *Peak IL-6 | 1.007 (1.001–1.014) | .018 | 1.007 (1.001–1.014) | .018 |
| CRP | 0.914 (0.640–1.305) | .620 | | |
| LDH | 0.999 (0.985–1.013) | .882 | | |

HR: hazards ratio; CI: confidence interval; BMI, body mass index; IPF, idiopathic pulmonary fibrosis; CTD, connective tissue disease-associated interstitial lung disease; UIP, usual interstitial pneumonia; FVC, forced vital capacity; DLco, diffusing capacity of the lungs for carbon monoxide; TLC, total lung capacity; SpO2, saturation of peripheral oxygen; PaO$_2$, partial pressure of oxygen; IL-6, interleukin-6; CRP, C-reactive protein; WBC, white blood cell; LDH, lactate dehydrogenase.

*Peak IL-6 –highest serum level of IL-6 among sequentially measured IL-6 during follow-up.

In our study, only high levels of IL-6 were an independent risk factor for mortality. A previous study supported our result. Among the 41 IPF patients, high levels of IL-6 were associated with mortality (OR 1.056, 95% CI 1.008–1.105, p = 0.021) [22]. In addition, in 67 patients with IPF (stable IPF 20 and AE IPF 47), Collard et al. showed that IL-6 was higher in the AE IPF group compared to the stable IPF group (10.1 pg/mL vs 5.3 pg/mL, p = 0.004). However, IL-6 was not associated with high mortality (OR 0.26, 95% CI: 0.06–1.24, p = 0.09). Our study results suggested that high levels of IL-6 are associated with the occurrence of AE resulting in high mortality. However, it is not clear whether high levels of IL-6 is a direct risk factor for mortality or predictor for the severity of AE because all deaths occurred in AE groups and groups with high levels of IL-6. Further studies are necessary to elucidate the relationship between degrees of IL-6 and the severity of AE suggesting high mortality.

This study has some limitations. First, it was a retrospective study conducted in a single center with a limited number of ILD patients. Also, the number of AE cases and deaths was too small to evaluate the risk factors for AE and deaths. However, the baseline characteristics of our subjects with the incidence and mortality rates were similar to those of patients in

previous reports. Second, the estimation of baseline oxygenation was performed by $SpO_2$ despite differences in arterial oxygen pressure or alveolar-arterial oxygen. Even though the level of $SpO_2$ was a statistically significant factor for predicting AE, $PaO_2$ was not included in our study. This is because $SpO_2$ was measured in most patients, but the data for oxygenation by arterial blood gas sampling was obtained only in a small number of patients. In addition, $SpO_2$ was well-validated for its correlation with $PaO_2$ on arterial blood gas analysis in a previous study [29]. Third, in this study, the sequential measurement of IL-6 may also be useful for predicting AE in the CTD-ILD group. However, the number of CTD-ILD patients was too small and the incidence of AE in patients with CTD-ILD was relatively high compared to previous studies [30]. Therefore, selection bias may exist in the CTD-group for the role of IL-6 in predicting AE. In the near future, studies with more CTD-ILD patients may be needed to evaluate the role of IL-6 as a prognostic factor in CTD-ILD patients.

## Conclusion

High serum levels of IL-6 during the follow-up period and lower baseline oxygenation were useful biomarkers for predicting AE. The optimal cut-off value of IL-6 was 25.20 pg/mL. In addition, high levels of serum IL-6 were a useful maker for predicting mortality.

## Supporting information

**S1 Fig. Comparison of survival curves with or without acute exacerbation groups.** (TIF)

## Author Contributions

**Conceptualization:** Jae Ha Lee, Chan Sun Park.

**Data curation:** Jae Ha Lee, Ji Hoon Jang, Hang-Jea Jang, Chan Sun Park, Sunggun Lee, Ji Yeon Kim.

**Formal analysis:** Ji Hoon Jang, Chan Sun Park.

**Funding acquisition:** Jae Ha Lee, Hyun Kuk Kim.

**Investigation:** Jin Han Park, Sunggun Lee.

**Methodology:** Jin Han Park, Hang-Jea Jang.

**Resources:** Hang-Jea Jang.

**Supervision:** Seong-Ho Kim, Hyun Kuk Kim.

**Validation:** Seong-Ho Kim.

**Visualization:** Jin Han Park, Sunggun Lee, Seong-Ho Kim, Ji Yeon Kim.

**Writing – original draft:** Jae Ha Lee, Ji Yeon Kim, Hyun Kuk Kim.

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
