## [Decision Letter · Decision Letter 0]

9 Apr 2021

PONE-D-21-04098

The Role of Interleukin-6 as a Prognostic Biomarker for Predicting Acute Exacerbation in Interstitial Lung Diseases

PLOS ONE

Dear Dr. Kim,

Thank you for submitting your manuscript to PLOS ONE. After careful consideration, we feel that it has merit but does not fully meet PLOS ONE’s publication criteria as it currently stands. Therefore, we invite you to submit a revised version of the manuscript that addresses the points raised during the review process.

Comments:

Table 1 & 2: Add % to the patient characteristic population ratio even explained at the bottom of the table, for example,  IPF, n (%)--- 57 (68.7).Figure 2: survival analysis, please correct mislabeling (IL6 level).As a predictive factor for AE in the CTD-group, n=3 is too small for biomarker study. I agree to the reviewer’s comments to remove CTD group from study.

     I hope reviewer's comments is helpful for improving manuscript.

We look forward to receiving your revised manuscript.

Kind regards,

Minghua Wu, M.D., Ph.D.

Academic Editor

PLOS ONE

PLOS requires an ORCID iD for the corresponding author in Editorial Manager on papers submitted after December 6th, 2016. Please ensure that you have an ORCID iD and that it is validated in Editorial Manager. To do this, go to ‘Update my Information’ (in the upper left-hand corner of the main menu), and click on the Fetch/Validate link next to the ORCID field. This will take you to the ORCID site and allow you to create a new iD or authenticate a pre-existing iD in Editorial Manager. Please see the following video for instructions on linking an ORCID iD to your Editorial Manager account: https://www.youtube.com/watch?v=_xcclfuvtxQ

Thank you for stating in the text of your manuscript " the requirement for written informed consent was waived due to the retrospective nature of this study". Please also add this information to your ethics statement in the online submission form."

Thank you for providing the date(s) when patient medical information was initially recorded. Please also include the date(s) on which your research team accessed the databases/records to obtain the retrospective data used in your study.

Thank you for stating the following in the Acknowledgments Section of your manuscript:

This work was supported by the 2019 Inje University Research Grant (20190041).

This study was supported by Inje university research grant.  The grant has nothing to do with the result of this research and is only to encourage the research.

Reviewers' comments:

Reviewer's Responses to Questions

**Comments to the Author**

1. Is the manuscript technically sound, and do the data support the conclusions?

Reviewer #1: Yes

Reviewer #2: No

2. Has the statistical analysis been performed appropriately and rigorously? 

Reviewer #1: Yes

Reviewer #2: No

3. Have the authors made all data underlying the findings in their manuscript fully available?

Reviewer #1: Yes

Reviewer #2: Yes

4. Is the manuscript presented in an intelligible fashion and written in standard English?

Reviewer #1: Yes

Reviewer #2: Yes

5. Review Comments to the Author

Reviewer #1: The manuscript by Dr. Lee et al demonstrated a Role of Interleukin-6 as a Prognostic Biomarker for Predicting Acute Exacerbation in Interstitial Lung Diseases. The authors followed up 83 patients who were diagnosed with ILD from 2016 to 2019 at the Haeundae Paik Hospital, Busan, South Korea. They measured the lung functional parameters and the serum levels of IL-6 at diagnosis with ILD and sequentially at follow-up visits. From the studies, the authors demonstrated that high levels of IL6 (OR 1.014, 95% CI: 1.001–1.027, p = 0.036) along with lower baseline saturations of peripheral oxygen (SpO2) were independent risk factors for AE. They also performed the receiver operating characteristic curve analysis, and found that the serum IL-6 to predict AE was 25.20 pg/mL with a sensitivity of 66.7% and specificity of 80.6%. Finally, the authors concluded that a high level of serum IL-6 is a useful biomarker to predict AE and poor prognosis in patients with ILD.

This manuscript demonstrated an important role of IL-6 as a biomarker for IPF-AE. Since the increased serum IL-6 and IL-8 level has already been reported (Papiris SA et al, Cytokine. 2018 Feb;102:168-172), the manuscript is somehow lack of novelty. However, the authors performed a longitudinal study to check IL-6 levels overtime. If the authors have already have the data, it will be interesting to show IL-6 levels overtime to see the dynamic changes of IL6 before the onset of the AE.

Minor comments.

Figure 2. Is the labels for “Peak IL-6 level>25.20” and “Peak IL-6 level<25.20” flipped?

Reviewer #2: Review of the manuscript: “The Role of Interleukin-6 as a Prognostic Biomarker for Predicting Acute Exacerbation in Interstitial Lung Diseases”

This retrospective study aimed to assess the merit of IL-6 to predict acute exacerbation (AE) of intestinal lung disease (ILD).

Comments

- Limitation: an important limitation of the paper is the low number of patients, with probable insufficient power (the power of the study sample needs to be determined) and the heterogeneity of included diseases Moreover, the OR and HR of IL6 to predict the occurrence of AE were low, close to 1, the question is their relevance in clinical pertinence

- Please precise the number of screened patients, those who fulfilled inclusion and exclusion criteria leading to the final number of recruited patients

- The ILD pattern on HRCT scan should me precised: UIP, NSIP…

- Please provide the detailed causes of AE. Please confirm that infections have been formally ruled out. It is important since infections may increase CRP and thus, IL-6 levels.

- Logistic regression analysis: please explain how were entered the different covariates in the model (stepwise, backward…). In this model, IL-6 was entered as a continuous variable; it would be interesting to consider IL-6 as a dichotomous variable (Lower/higher)

- Multivariate analyses: gender, disease duration, and HRCT scan pattern should be included in the models

- The pertinence to include in the models both CRP and IL-6 is questionable given the strong relationship between these 2 variables

- Kaplan-Meier survival curve: a table with the number of patients should be added below the figure

- Given the very low number of patients with CTD-ILD, and the low frequency of AE, the reviewer suggests the deletion of the sub analysis of the CTD-group

- How were Handled missing data?

6. PLOS authors have the option to publish the peer review history of their article (what does this mean?). If published, this will include your full peer review and any attached files.

Reviewer #1: No

Reviewer #2: No

---

## [Author Response · Author response to Decision Letter 0]

28 May 2021

Dear Editor,

We would like to thank you for the opportunity to resubmit a revised copy of our manuscript. We would also like to take this opportunity to express our thanks to the reviewers for the positive feedback and helpful comments for correction or modification.

We have made substantial changes to the manuscript according to the reviewers’ comments. The changes made are marked in red in the revised manuscript. Our point-by-point responses to the reviewers' comments are provided below. We hope the revised manuscript is suitable for publication in PLOS ONE.

[Response to Academic Editor's comments]

Q1. Table 1 & 2: Add % to the patient characteristic population ratio even explained at the bottom of the table, for example, IPF, n (%)--- 57 (68.7).

R1. Thank you for pointing this out. Based on this comment, we add % in the Table 1 & 2.

Q2. Figure 2: survival analysis, please correct mislabeling (IL6 level).

R1. Thank you for your comment. Sorry for our mistake. This is a typing error, and was corrected.

Q3. As a predictive factor for AE in the CTD-group, n=3 is too small for biomarker study. I agree to the reviewer’s comments to remove CTD group from study.

R1. Thank you for pointing this out. We agree to your comment. The paragraph “Predictive factor for AE in the CTD-group” removed from the manuscript. 

[Journal Requirements]

Q1. Please ensure that your manuscript meets PLOS ONE's style requirements, including those for file naming.

R1. Thank you for pointing this out. Based on this comment, we ensured that our manuscript meets PLOS ONE's style requirements. 

Q2. PLOS requires an ORCID iD for the corresponding author in Editorial Manager on papers submitted after December 6th, 2016. Please ensure that you have an ORCID iD and that it is validated in Editorial Manager. To do this, go to ‘Update my Information’ (in the upper left-hand corner of the main menu), and click on the Fetch/Validate link next to the ORCID field. This will take you to the ORCID site and allow you to create a new iD or authenticate a pre-existing iD in Editorial Manager.

R2. Thank you for pointing this out. We updated an ORCID iD for the corresponding author in Editorial Manager. 

Q3. Thank you for stating in the text of your manuscript "the requirement for written informed consent was waived due to the retrospective nature of this study". Please also add this information to your ethics statement in the online submission form."

R3. Thank you for pointing this out. We add the statement "the requirement for written informed consent was waived due to the retrospective nature of this study" to the ethic statement in the online submission form. 

Q4. Thank you for providing the date(s) when patient medical information was initially recorded. Please also include the date(s) on which your research team accessed the databases/records to obtain the retrospective data used in your study.

R4. Thank you for pointing this out. We added the date in medical records in Clinical information of MATERIALS AND METHODS as below. 

[MATERIALS AND METHODS, Clinical information]

Clinical data were retrospectively obtained from the medical records at January 3, 2020.

Q5. Thank you for stating the following in the Acknowledgments Section of your manuscript:

This work was supported by the 2019 Inje University Research Grant (20190041). We note that you have provided funding information that is not currently declared in your Funding Statement. However, funding information should not appear in the Acknowledgments section or other areas of your manuscript. We will only publish funding information present in the Funding Statement section of the online submission form.

This study was supported by Inje university research grant. The grant has nothing to do with the result of this research and is only to encourage the research.

R5. Thank you for pointing this out. We deleted this sentence in the Acknowledgments and also inserted the sentence at the end of the cover letter.

[Response to Reviewer 1's comments]

Q1. Figure 2. Is the labels for “Peak IL-6 level>25.20” and “Peak IL-6 level<25.20” flipped?

R1. Thank you for your comment. Sorry for our mistake. This is a typing error, and was corrected.

[Response to Reviewer 2's comments]

Q1. Limitation: an important limitation of the paper is the low number of patients, with probable insufficient power (the power of the study sample needs to be determined) and the heterogeneity of included diseases Moreover, the OR and HR of IL6 to predict the occurrence of AE were low, close to 1, the question is their relevance in clinical pertinence

R1. Thank you for pointing this out. I agree with your opinion. In this study, the value of OR of IL-6 to predict to acute exacerbation and mortality was 1.014 (95% CI: 1.001 – 1.027) and 1.007 (95%CI: 1.001 – 1.014), respectively. However, the P–value was statistically significant. The diagnosis of acute exacerbation is still difficult and very important for good prognosis. Although revised diagnostic criteria of AE IPF, early and accurate diagnosis of AE ILD is still big challenging. In our county, other biomarker including KL-6 and SP-D had not been available. Therefore, we evaluated the role of IL-6 to predict AE. As you know, there cannot be a perfect biomarker. We expect that IL-6 might be helpful to predict to AE in addition to other method. Also IL-6 can be more useful as an acute reactant than other biomarker. 

Q2. Please precise the number of screened patients, those who fulfilled inclusion and exclusion criteria leading to the final number of recruited patients

R2. Thank you for pointing this out. According to reviewer’s comment, we added flow-chart for enrollment (Fig 1) in Study population of RESUTLS as below

[Study population in RESULTS]

From December 2016 to September 2019, 405 patients with ILD at Haeundae Paik hospital (Busan, Republic of Korea) were screening for this study. Among them, patients who did not perform IL-6 test or follow up at IL-6 test at least 6 months were excluded (Fig 1). 

Q3. The ILD pattern on HRCT scan should me precised: UIP, NSIP…

R3. Thank you for pointing this out. We added Radiologic pattern on HRCT to the table 1.

Q4. Please provide the detailed causes of AE. Please confirm that infections have been formally ruled out. It is important since infections may increase CRP and thus, IL-6 levels.

R4. Thank you for pointing this out. As a result of re-analysis, 11 patients and 4 patients were reclassified as idiopathic and triggered AE, respectively. Triggered AE was defined as proven pathogen or increased level of procalcitonin. We added detail data in Predictive factors in RESULTS as below. 

[RESULTS, Predictive factors]

Among the 15 (18.1%) patients with AE (idiopathic AE: 11 patients and triggered AE: 4 patients), IPF patients were most common (80%) while the other patients were classified with CTD (20%).

Q5. Logistic regression analysis: please explain how were entered the different covariates in the model (stepwise, backward…). In this model, IL-6 was entered as a continuous variable; it would be interesting to consider IL-6 as a dichotomous variable (Lower/higher)

R5. Thank your comment. We modified the sentence in Statistical analysis as below. 

[MATERIALS AND METHODS, Statistical analysis]

Multivariate analyses, using Cox regression with backward and stepwise elimination, were performed in order to identify prognostic factors which are independently related to mortality.

Q6. Multivariate analyses: gender, disease duration, and HRCT scan pattern should be included in the models

R6. Thank you for comment. According to comment, we added the variables including gender, disease duration and radiologic pattern to the table 3 and 4. However, NSIP, OP and indeterminate pattern were excluded from multivariate analysis because they caused statistical error. 

Q7. The pertinence to include in the models both CRP and IL-6 is questionable given the strong relationship between these 2 variables

R7. Thank you for your comment. CRP is produced by hepatocyte in the liver in response to stimulation of pro-inflammatory cytokines. Therefore, CRP has been used as biomarker for inflammation. In several study, elevated level of CRP was reported as useful biomarker to predict AE ILD. In this study, baseline level of CRP was significantly different between AE group and non-AE group. However, high level of CRP was not useful to predict AE and mortality, although IL-6 was useful to predict both. In this study, the role of CRP and IL-6 was different to predict AE and mortality. In the field of ILD, we thought the role of CRP and IL-6 are different and independent. 

Q8. Kaplan-Meier survival curve: a table with the number of patients should be added below the figure

R8. Thank you for your comment. We added a table with the number of patients in Figure 3.

Q9. Given the very low number of patients with CTD-ILD, and the low frequency of AE, the reviewer suggests the deletion of the sub analysis of the CTD-group

R9. Thank you for your comment. We agree your opinion. The paragraph “Predictive factor for AE in the CTD-group” was removed from the manuscript. 

Q10. How were Handled missing data?

R10. Thank you for comment. There was some missing data. Therefore, the missing data was null out and the remaining data was analyzed. Variables with many missing data such as RVSP, were excluded from the statistical analysis because they caused statistical error. 

Sincerely,

Hyun Kuk Kim, M.D., Ph.D.

Department of Pulmonary and Critical Care Medicine, Inje University Haeundae Paik Hospital, Inje University College of Medicine, Haeundae-ro 875, Haeundae-gu, 48108, Busan, Republic of Korea.

---

## [Decision Letter · Decision Letter 1]

15 Jul 2021

The Role of Interleukin-6 as a Prognostic Biomarker for Predicting Acute Exacerbation in Interstitial Lung Diseases

PONE-D-21-04098R1

Dear Dr. Kim,

We’re pleased to inform you that your manuscript has been judged scientifically suitable for publication and will be formally accepted for publication once it meets all outstanding technical requirements.

Kind regards,

Minghua Wu, M.D., Ph.D.

Academic Editor

PLOS ONE

Additional Editor Comments (optional):

Reviewers' comments:

Reviewer's Responses to Questions

**Comments to the Author**

1. If the authors have adequately addressed your comments raised in a previous round of review and you feel that this manuscript is now acceptable for publication, you may indicate that here to bypass the “Comments to the Author” section, enter your conflict of interest statement in the “Confidential to Editor” section, and submit your "Accept" recommendation.

Reviewer #1: All comments have been addressed

Reviewer #2: All comments have been addressed

2. Is the manuscript technically sound, and do the data support the conclusions?

Reviewer #1: Yes

Reviewer #2: Yes

3. Has the statistical analysis been performed appropriately and rigorously? 

Reviewer #1: Yes

Reviewer #2: Yes

4. Have the authors made all data underlying the findings in their manuscript fully available?

Reviewer #1: Yes

Reviewer #2: Yes

5. Is the manuscript presented in an intelligible fashion and written in standard English?

Reviewer #1: Yes

Reviewer #2: Yes

6. Review Comments to the Author

Reviewer #1: I would like to thank the authors for addressing my questions! I have no other concerns and comments.

Reviewer #2: The authors have adequately answered to all the raised comments. The manuscript has been significantly improved.

7. PLOS authors have the option to publish the peer review history of their article (what does this mean?). If published, this will include your full peer review and any attached files.

Reviewer #1: No

Reviewer #2: No

---

## [Editor Report · Acceptance letter]

19 Jul 2021

PONE-D-21-04098R1 

The Role of Interleukin-6 as a Prognostic Biomarker for Predicting Acute Exacerbation in Interstitial Lung Diseases 

Dear Dr. Kim:

I'm pleased to inform you that your manuscript has been deemed suitable for publication in PLOS ONE. Congratulations! Your manuscript is now with our production department. 

Kind regards, 

on behalf of

Dr. Minghua Wu 

Academic Editor

PLOS ONE